# Induced Autolysis of Engineered Yeast Residue as a Means to Simplify Downstream Processing for Valorization—A Case Study

Joana F. Fundo *, Teresa Deuchande, Daniela A. Rodrigues, Lígia L. Pimentel, Susana S. M. P. Vidigal, Luís M. Rodríguez-Alcalá, Manuela E. Pintado and Ana L. Amaro

CBQF—Centro de Biotecnologia e Química Fina—Laboratório Associado, Escola Superior de Biotecnologia, Universidade Católica Portuguesa, Rua Diogo Botelho 1327, 4169-005 Porto, Portugal; tdeuchande@ucp.pt (T.D.); darodrigues@ucp.pt (D.A.R.); lpimentel@ucp.pt (L.L.P.); svidigal@ucp.pt (S.S.M.P.V.); lalcala@ucp.pt (L.M.R.-A.); mpintado@ucp.pt (M.E.P.); aamaro@ucp.pt (A.L.A.)
* Correspondence: jfundo@ucp.pt; Tel.: +351-22-558-00-00

**Abstract:** The objective of this work was to study the efficiency of different autolysis processes, combining different temperatures and pH conditions, when applied to a genetically engineered yeast residue. The determination of the supernatants' dry weight showed that the autolysis time could be reduced to half, from 4 to 2 h, if the residue pH was increased from 5 to 8 at 50 °C (18.20% for 4 h and 18.70% for 2 h with a higher pH). This result allowed us to select a short autolysis time to proceed with the second part of the experiments. The application of this faster induced autolysis process enabled us to obtain supernatants with higher concentrations of relevant compounds, such as some amino acids and minerals. An increase in leucine (of around 7%), aspartic acid, valine, phenylalanine, isoleucine and serine (approximately 2%) was observed in the autolyzed samples, when compared to the untreated ones. Also, regarding minerals, the autolysis process allowed us to obtain significantly higher amounts of potassium in the treated samples' supernatants. This work allowed the selection of a fast and low-cost induced autolysis process for synthetic biotechnology-derived spent yeast residue to attain a product rich in high-value compounds, which can be used in commercial applications, for example, as an animal feed additive.

**Keywords:** synthetic biotechnology; spent yeast residue; nutrient source; autolysis; physicochemical composition; animal feed

## 1. Introduction

Spent yeast residue (SYR) is one of the major by-products generated by fermentation-based industries, such as brewing, baking or wine production. The brewing industry stands out with an annual spent brewer's yeast production of around 400 thousand tons [1]. For fermentation-based industries, the production of huge quantities of this residue represents a management challenge from both ecological and economical points of view. *Saccharomyces cerevisiae* is, by far, the most commonly used yeast in this type of industry, representing a powerful and versatile industrial tool for multiple purposes [2]. Its industrial use is mainly due to its fast growth, good ethanol-producing capacity and great tolerance against environmental stress, including high ethanol concentrations and low oxygen levels [3,4].

Currently, SYR also represents a challenging problem for the synthetic biotechnology industry. This industry conducts precision fermentations to produce complex and valuable biomolecules for applications in the cosmetics, pharmaceutical and agri-food industries. These fermentations are driven by genetically modified yeasts, mostly *S. cerevisiae* strains, which are used as cell factories to produce valuable molecules. This industry is under expansion, and an increasing number of synthetic biotechnology plants making use of innovative and

sustainable processes to promote the conversion of plant sugars into stable, alternative and high-commercial-interest biomolecules is foreseen [5].

Spent yeast residue represents a cheap, attractive and easily available source of valuable molecules and bioactive compounds [6], which is suitable for different valorization strategies. The valorization of SYR can be achieved through the extraction of high-value components, such as proteins, polysaccharides, fibers, flavor compounds and phytochemicals, which can be reused as nutritionally, pharmacologically and cosmetically functional ingredients [7].

Regarding the applications of this waste by-product, the literature reports many possibilities of application in very distinct areas, unequivocally showing its versatility and valorization potential. Due to its high nutritional value and low cost, this residue can be used as a fermentation subtract and/or additive [7], as a biosorption element [8], and as a food ingredient or nutraceutical due to the presence of high levels of polyphenolic compounds [7]. Spent yeast extracts are also known to be rich sources of proteins, which can be incorporated as supplements in animal diets, namely for fish and ruminants [9–11]. Its incorporation in animal feed is currently one of the main strategies for reusing this by-product, and spent brewer' yeast, in particular, has long been incorporated in ruminant diets as a protein additive. It has been demonstrated that, under in vitro conditions, spent brewer's yeast from craft beer, which contains antimicrobial α- and β-acids, can prevent excessive rumen protein degradation by rumen hyper-ammonia-producing bacteria when used as a protein additive [7,11].

However, the use of the majority of spent yeast components requires yeast cell lysis, commonly known as autolysis. For valorization strategies, the development of an appropriate spent yeast residue cell disruption process is of the utmost importance for an efficient and cost-effective recovery of target compounds [6]. Although a naturally occurring event, autolysis can be induced and enhanced by exposing spent yeast residue to elevated temperatures, organic solvents or physical processes, such as ultrasounds or high pressure [12]. Depending on the final valorization objective, it is possible to choose the most appropriate autolysis process or even a combination of processes. The costs associated with this procedure are an important issue that needs to be considered; furthermore, due to the focus on waste recovery strategies, the target cost has to be as low as possible so that its application is worthwhile.

Several research studies have focused on induced autolysis to obtain additives/products with a relevant nutritional composition from SYR derived from beer fermentation. However, to the best of our knowledge, there have been no studies on the optimization of an efficient autolysis process for SYR derived from these platforms in which genetically modified yeasts strains are used. Based on previous studies on the characterization of such residues, it is possible to observe that SYRs derived from the production of different types of beer have significant compositional differences. Thus, it is important to assess if the autolysis methods currently applied to spent brewer's yeast are effective for the extraction of valuable compounds from these new waste streams, or if other methods should be developed for this purpose.

Therefore, as this study aimed to facilitate downstream valorization strategies for a synthetic-biology-derived SYR, a brief review of the methods used to autolyze yeast cells from spent brewer's yeast is given, emphasizing the efficiency, the cost and the process rate of these methods. An overview of the suitability of processed residue for application as a supplement for animal feeding is also discussed, considering the characteristics of existing animal feed supplements that are commercially available.

## 2. Yeast Cell Disruption Strategies—An Overview

The main objective of induced autolysis is to rupture yeast cell wall and release cell-soluble constituents, which are predominantly compounds of potential biological interest [13]. As mentioned above, this process is of the utmost importance to the success of a defined SYR valorization strategy and should be designed accordingly.

If the economic issues related to the induced autolysis process are not to be considered, the major difference between several cell disruption procedures is the size of the cell fragments generated and the method by which cells are sheared. Even though several mechanical methods (bead mills, high-pressure homogenization and ultrasonication), which are superior in terms of product recovery, are industrially preferred, their poor selectivity and complicated downstream processing are the major drawbacks. Non-mechanical methods are mild and result in large cell fragments, thus promoting downstream process operations. However, their limited recovery efficiency and expensiveness restrict their general applicability [6].

Nevertheless, if induced autolysis with temperature is combined with, for example, the addition of chemical compounds (such as sodium chloride and ethyl acetate), the disruption effectiveness is higher (around 98%) than some mechanical methods, namely bead mill and ultrasonication (80%), and leads to a higher cleavage of amino acids from cell protein (307, 155 and 115 mg g$^{-1}$ of yeast when using autolysis with temperature/chemical agents, ultrasonication and bead mill, respectively) [7].

Particularly, if spent yeast is to be used as an additive for animal feed, the autolysis process plays an important role. In this case, disrupting yeast cell walls before feeding improves the availability of intracellular nutrients and facilitates their digestion and absorption [7,14]. In order to turn the incorporation of this additive in animal feeding profitable, the transformation process has to be as simple, fast, efficient and economical as possible.

Table 1 shows a brief summary of some of the published works that are relevant for an induced autolysis design. The different autolysis conditions and different methodologies used are described, and the main takeaways are highlighted.

**Table 1.** Description of the different conditions and methodologies used in previous research, and the main takeaways for induced autolysis of wild-type yeasts.

| Strain | Autolysis Conditions/Method | Main Takeaways | Ref. |
|---|---|---|---|
| *S. cerevisiae* | -Pulse electric field (PEF): 5–20 kV/cm, 1–2000 pulses, and 15 µs pulse width. -Autolysis: 52 °C/72 h/ pH 5.5. | -PEF increased the final amino acid and total solid contents. -PEF was found to accelerate the progress of autolysis (up to 78%). | [15] |
| | -Autolysis: various pH values (4.0, 5.5, 7.0 and 8.5) and use of chemical autolysis promoters (ethyl acetate and chitosan). | -Best combination—pH 5.5 and ethyl acetate. -Good peptidase activities at these pH values. -Yeast extract with higher turbidity when produced at pH 7.0 and 8.5. | [16] |
| | -Influence of temperature, pH and ethanol concentration on PEF-induced autolysis. | -At the same incubation time, the amount of mannose released from PEF-treated cells ranged from 80 mg L$^{-1}$, when incubated with 25% ethanol, to 190 mg L$^{-1}$, when incubated at 43 °C. | [17] |
| | -Scale-up ultrasonic disruption of yeast (Barbell Horn Ultrasonic Technology—BHUT), a usual method for lab scale. | -BHUT can be successfully used on a large scale. -BHUT-based equipment allows efficient extraction of total protein and alkaline phosphatase from yeast cells. | [18] |
| | -Influence of ultrasound intensity, sonication time, temperature and yeast concentrations. -2 probe depths; ionic strengths at 0.05, 0.55 and 1.05 M; and levels of ethanol addition at 10, 50 and 100 mM. | -Release of polysaccharides and proteins was affected by most of the processing parameters. -The parameter, temperature, had the greatest influence on selectivity of released product. | [19] |
| | -Effect of temperature (45, 50, 55 and 60 °C) and reaction time (ranging from 8 to 72 h). | -Optimum temperature/time combination: 50 °C for 24 h, on the basis of α-amino nitrogen and the protein contents. - Also favorable for sensory analysis. | [20] |
| | -Effect of high pressure (HPH) (200 to 600 MPa) for 0 to 120 min. -Activity of the vacuolar proteases was monitored during the autolysis. The autolytic capacity of yeast was determined based on the physicochemical characteristics of the yeast extract. | -At 200 and 400 MPa, the proteolytic activity was enhanced up to 160% after 40 and 10 min, respectively. -Autolysis was significantly accelerated, in combination with cellular permeabilization, when achieved with HP treatment. -At 600 MPa, proteolytic enzymes were gradually inactivated, leading to the inhibition of autolysis. | [21] |
| | -Comparing conventional methods (autolysis and mechanical rupture) with enzymatic hydrolysis using proteolytic enzymes. | -The hydrolysate produced at pH of 5.5, 100% substrate, 10% enzyme/substrate ratio and 60 °C resulted in a maximized yield with enhanced antioxidant properties. -Enzymatic hydrolysis promoted more efficient release of solids, proteins and cell walls. | [22] |

**Table 1.** *Cont.*

| Strain | Autolysis Conditions/Method | Main Takeways | Ref. |
|---|---|---|---|
| *S. pastorianus* | -Mechanical disruption.<br>-Separation of the β-glucan-rich fraction.<br>-Extract rich in native proteins and enzymes. | -The best autolysis conditions were 36 °C/6 h | [23] |
| *S. bayanus* | -High-pressure homogenization (HPH) at 5, 100 and 150 MPa and comparison with thermolysis (121 °C/2 h). | -HPH seemed to be a promising technique (150 MPa was the best operation condition).<br>-Thermolysis was more efficient. | [24] |
| *S. cerevisiae/ Sacch. uvarum* | Study on autolytic release of polysaccharides from cell walls, in a model medium, over a nine-month period of ageing over lees, and the effect of adding β-glucanase. | -The addition of enzyme promoted complete autolysis in less time (2–3 weeks instead 5 months)<br>-Enzyme-assisted autolysis promoted the production of smaller-molecular-weight fragments.<br>-The extension of autolysis was different for different strains. | [25] |
| *Kluyveromyces fragilis* | -NaCl-induced autolysis studied as a function of time ($t$), at different initial yeast concentrations ($X_0$) and reaction temperatures ($T$) | -Protein solubilization was temperature dependent.<br>-Hydrolysis of total carbohydrates was found to be controlled firstly by yeast concentration and secondly by temperature. | [26] |

This targeted literature review allows us to verify the use of *S. cerevisiae* as a key model strain for autolysis studies. As already stated, *S. cerevisiae* is undoubtedly the most known, studied and used yeast species. *S. cerevisiae*'s great adaptability and growth capacity makes it a model organism in eukaryotic biology and the first eukaryotic organism to have its genome sequenced [2].

As stated before, it is important to verify the efficacy of the autolysis process since synthetic biotechnology-derived SYRs may present a different chemical composition when compared to SYRs from native *S. cerevisiae.* Thus, a case study is herein presented to confirm if compositional differences in SYRs imply changes in the efficiency of the autolysis process and differences in the compositional profile of the generated products.

## 3. Case Study—Research Methodology and Main Results and Conclusions

Based on the brief literature review presented in the previous section, and considering the need to identify an effective, economical, fast and easy-to-carry-out autolysis process to be applied to a synthetic biotechnology-derived SYR, an induced autolysis process using different combinations of temperature, pH and time was studied. The use of different enzymes was also considered, as well as the use of some mechanical methods, namely ultrasounds and high-pressure homogenization. Even if these mechanical methods are, as reported in the literature, more efficient than autolysis based on temperature and pH, their incorporation into the process turns the industrial costs restrictive.

### 3.1. Samples

A SYR derived from a fermentation process driven by a *S. cerevisiae* strain and modified to produce β-farnesene, which was provided by Amyris, Inc. (Emeryville, CA, USA), was autolyzed under different conditions of pH, temperature and time.

Two different pH levels, 5.5 (the residue's pH at arrival) and 8, were studied. The higher pH value was attained using NaOH (Sigma, Aldrich, St. Louis, MO, USA) at 3 M. The autolysis process was implemented at two different controlled temperatures, 50 and 70 °C, using a water bath. The samples were collected for analysis at different treatment times (2, 4, 6, 8 and 24 h). The induced autolysis process was performed in triplicate in sterilized 500 mL glass jars using batches of 250 mL of whole spent yeast.

In the initial stage of the experiment, some tests were also carried out after the addition of different enzymes, namely alcalase (Sigma, Aldrich) at two different concentrations (1 and 3%) and a mixture of alcalase and celluclast (Sigma, Aldrich; both at 3%).

### 3.2. Methodology

3.2.1. Determination of the Best Combination of pH, Time and Temperature for Induced Autolysis

The first objective of this study was to select the most promising combination of temperature, pH and autolysis time, in order to achieve effective cellular breakdown while

maintaining the process costs low. For this, the SYR supernatants' dry weight and protein content were selected as indicators of autolysis efficiency. The protein content is one of the main indicators of autolyzed spent yeast's suitability for animal feed supplementation [27]. To obtain the supernatants, the non-autolyzed and autolyzed spent yeast suspensions were centrifuged at 5000 rpm for 10 min, and the supernatants were collected for the dry weigh and protein content analysis. These determinations allow a prompt, easy and effective evaluation of the autolysis conditions. A brief comparison with the samples autolyzed with the addition of enzymes was also performed.

### 3.2.2. Comparison of Nutritional Composition of Non-Autolyzed and Autolyzed SYR

In the second stage of the experimental work, the SYR samples were autolyzed according to the conditions previously selected (pH and temperature), and the resulting product was compared with the non-autolyzed samples. In addition, the supernatants of the autolyzed and non-autolyzed SYR samples were also evaluated in terms of their nutritional characteristics [13]. These samples were subjected to different compositional analysis, namely dry weight, mineral content, total protein content, amino acid profile, and total sugar and total lipid contents and their respective profiles.

The samples' dry weight was determined using an oven at 105 °C for 24 h, according to the Association of Official Analytical Chemists (AOAC) [28]. For this analysis, 1 mL of the autolyzed and non-autolyzed supernatants and the whole spent yeast samples was used, and the analysis was carried out in triplicate.

The total protein content of SYR (whole sample and supernatants) was determined using the Dumas method. This method is based on the combustion of the whole sample in an oxygen-enriched atmosphere at a high temperature to ensure complete combustion [29]. A Dumatec$^{TM}$ 8000 (Foss, Hilleroed, Denmark) was used.

The amino acid profile for the whole SYR samples (total amino acids) was determined through an acid hydrolysis of the samples according to [30]. Briefly, 10 mg of each sample was used and 3 mL of HCL (6 M) was added; then, the mixture was vortexed and flushed with nitrogen. The samples were then incubated for 20 h at 115 °C and diluted with 4 mL of water. The pH value was adjusted to 3.5 and deionized water was added until a final volume of 10 mL. The samples' supernatants were used directly, only diluted in 0.1 M of HCl. Both samples (whole and supernatants) were filtered using 0.45 μm filters to be analysed using high-performance liquid chromatography (HPLC). The amino acid profile was achieved using an HPLC equipped with a Chromolith$^®$ Performance RP18 (4.6 mm × 100 mm) column from Merck, a high-resolution fluorescence detector, and an "autosampler".

The quantification of total sugars and the determination of sugar profiles were performed through polysaccharide reduction and acetylation in alditol acetates. Before the derivatization reaction, polysaccharides were hydrolyzed and then separated and detected via GC-FID and quantified using 2-deoxyglucose as the internal standard [31,32]. Briefly, 2 mg of dried whole and supernatant samples was hydrolyzed with 200 μL of $H_2SO_4$ (58% *v/v*) for 3 h at room temperature and then with 1 M $H_2SO_4$ at 100 °C for 2.5 h. After this, reduction and acetylation of the hydrolyzed samples was carried out with the addition of the mixture of the internal standard at 2 mg mL$^{-1}$ with 25% $NH_3$ and 15% $NaBH_4$ prepared in 3 M $NH_3$. The mixture was incubated at 30 °C for 1 h. Then, two washes with glacial acetic acid were performed, and 1-methylimidazole and acetic anhydride were added. After this, the tubes were kept at 30 °C for 30 min, and the organic phase was washed twice with 3.0 mL of distilled water, 2.5 mL of dichloromethane, and, then again, twice with 3 mL of distilled water. The organic phase was evaporated, and then anhydrous acetone was added and evaporated twice. Immediately before the GC-FID analysis, the dried sugars were suspended in 100 μL of anhydrous acetone.

For the extraction and quantification of total lipids, a mixture of solvents with different polarities was used since yeasts contain lipids with both polar and apolar properties. The extracted lipids were then purified via phase separation and gravimetrically quantified

according to [33]. The lipid profiling was carried out using HPLC-ELSD as described by [34]. Briefly, the samples were weighted and dissolved in dichloromethane to a concentration of 3 mg mL$^{-1}$. Afterward, the samples were analyzed using an HPLC (model 1260 Infinity II; Agilent Technologies, Santa Clara, CA, USA) attached to an evaporative light scattering Detector (ELSD; 1290 Infinity II, Agilent Technologies, Santa Clara, CA, USA), using nitrogen as the nebulizing gas, coupled to a Zorbax RX-SIL column (2.1 mm × 150 mm, 5 μm; Agilent Technologies, Santa Clara, CA, USA). The analysis conditions were assayed as described by [35] with slight changes. The composition of the mobile phases was as follows: A, isooctane/ethyl acetate (99.8:0.2, *v/v*); B, acetone/ethyl acetate (2:1, *v/v*) containing 0.1% acetic acid (*v/v*); C, 2-propanol/water (85:15, *v/v*) containing 0.013% acetic acid (*v/v*) and 0.031% of TEA *v/v*; and D, EtAc. The flow rate was set at 0.275 mL min$^{-1}$ with an injection volume of 20 μL. The detector was set as follows: evaporator and nebulizer temperature at 60 °C with nitrogen as the nebulizing gas at 1.20 SLM flow rate. For the determination of the elution order, the pure standards were injected, as well as available bibliography was used [36]. All samples were injected at least in triplicate.

The concentrations of minerals were determined using an optical emission spectrometer, Model Optima 7000 DV™ ICP-OES (Dual View, PerkinElmer Life and Analytical Sciences, Shelton, CT, USA), with a radial configuration, as described by [37]. Prior to the analysis, the autolyzed and non-autolyzed supernatants (2 mL) were mixed with 5 mL of HNO$_3$ (65%) and 1 mL of H$_2$O$_2$ (30%) in an appropriate vessel and digested in a microwave system (Speedwave MWS-3+, Berghof, Eningen, Germany), following an established program of times/temperatures. This analysis was carried out in triplicate and the result was expressed in mg L$^{-1}$.

Statistically significant differences between all samples, in relation to dry weigh, total protein, and total sugar and total lipid contents, were determined using an unpaired *t*-test for each parameter individually and assuming equal variances. In relation to amino acids, as well as sugar and lipid profiles, multiple *t*-tests corrected for multiple comparisons using the Holm–Sidak method for *p* = 0.05 were conducted, assuming consistent variances. These statistical analyses were performed using GraphPad Prism version 8.0.2 (San Diego, CA, USA). Hierarchical clustering analyses (HCA) were carried out using MetaboAnalyst 5.0 (McGill University, Canada). For the HCA of the whole samples before and after induced autolysis, a matrix including 38 variables (compositional data) and 6 observations (3 replicates of each condition) was constructed, whereas for the analysis of the supernatants of whole and autolyzed spent yeast, a data matrix consisting of 32 variables (compositional data) and 6 observations (3 replicates of each condition) was constructed. In both cases, the data were auto scaled before analysis, and the HCA was performed using the Euclidean distance measure and the ward's algorithm. The results were presented in the form of a heatmap.

## 4. Results and Discussion

### 4.1. Determination of the Best Combination of pH, Time and Temperature for Induced Autolysis

In Table 2, it is possible to compare the dry weight and protein yields (calculated using the day 0 values as a reference) in the supernatants of the SYR samples after being treated with different combinations of pH, temperature, enzyme addition and time.

Autolysis induced for 2 h at a lower temperature (50 °C) resulted in the highest dry weight yield (18.70%), even when compared with autolysis with the addition of enzymes (around 10%). According to the results reported by Tanguler et al. for spent brewer´s yeast [20], yields obtained at lower temperatures were higher than yields achieved when higher temperatures were used. In addition, Suphanthrika et al. [38] stated that autolysis between 45 and 50 °C led to a maximum yield with respect to the amount of solids released into the liquid yeast extract from baker's yeast.

**Table 2.** SYR supernatants' dry weight and protein contents after different induced autolysis conditions. The results are presented as the normalized values related to day 0 ($D_0$) of storage.

| Treatment | Induced Autolysis Time (H) | | | |
|---|---|---|---|---|
| | Dry Weight (% $D_0$ Value) | | Protein (% $D_0$ Value) | |
| | 2 | 4 | 2 | 4 |
| pH 5.5; T 70 °C | 8.20 | 18.20 | 4.4 | 13.2 |
| pH 8.0; T 50 °C | 18.70 | 19.30 | 4.4 | 15.4 |
| pH 8.0; T 70 °C | * | * | 3.7 | 5.0 |
| Alcalase 1% | 10.20 | 34.35 | 2.2 | * |
| Alcalase 3% | 10.60 | 38.20 | 2.7 | * |
| Enzyme Mix | 6.90 | 15.00 | 3.8 | 13.1 |

* Value lower than the one obtained on day 0.

The increase in the samples' pH with the addition of NaOH also promoted this increase in autolysis efficacy when applied to spent yeast from beer manufacture [39].

When the autolysis time was extended to 4 h, the addition of alcalase was, without any doubt, the most efficient process with respect to dry weight yields (more than 30%). However, the addition of a mixture of enzymes (alcalase and celluclast) did not prove to be effective in increasing the efficiency of autolysis, since the values for dry weight, both at time 2 and 4, were lower when compared with the other treatments.

Protein yield varied with autolysis time. Although an increase of 2 h in the treatment resulted in an increase in protein yield by almost 10%, the 4 h treatment carried out at a lower temperature showed the best yield. These results are in accordance with Suphanthrika et al. [38] who reported the lowest protein yields at higher temperatures (55 and 60 °C) for the production of a baker's yeast extract.

Based on these results and with the objective of applying an autolysis process that was simultaneously fast, effective and economically viable, we decided to combine a time of 2 h and the addition of NaOH to increase the pH to 8 with a temperature of 50 °C.

*4.2. Compositional Comparison of Non-Autolyzed and Autolyzed Samples*

4.2.1. Dry Weight

The dry weight of the SYR samples was determined and compared, as shown in Table 3. As expected, no differences were attained for this parameter when the whole spent yeast samples were evaluated. However, with respect to the samples' supernatants, significant differences were observed between the untreated and the autolyzed samples.

**Table 3.** Dry weight of spent yeast residue samples. Results are expressed as mean $\pm$ standard deviation. Significant differences are represented by different letters in the same rows for $p < 0.05$. Lowercase letters refer to the whole spent yeast samples and uppercase letters refer to the supernatant samples.

| Sample | Dry Weight (%) |
|---|---|
| Untreated whole spent yeast | 17.48 $\pm$ 0.21 [a] |
| Autolyzed whole spent yeast | 17.14 $\pm$ 0.03 [a] |
| Untreated supernatant | 7.27 $\pm$ 0.28 [A] |
| Autolyzed supernatant | 9.37 $\pm$ 0.00 [B] |

The supernatants of the autolyzed samples showed higher values of dry weight, which suggests higher amounts of solid components [40] resulting from the release of intracellular constituents. This could be due to the amounts of free amino acids, peptides, sugars and nucleotides that became available through induced autolysis [41,42].

4.2.2. General Nutritional Composition: Proteins, Sugars and Lipid Contents

The nutritional composition of the SYR samples, regarding proteins, sugars, and lipid contents, was determined, and the results are presented in Table 4. It was possible to verify some differences when comparing the general nutritional composition of the SYR used in this study with the native spent brewer's yeast. Generally, and using values reported in the literature, it is possible to conclude that the residue used in this study had slightly lower levels of proteins (in some cases, less than 23.5%) but higher percentages of total sugars and lipids (more than 1.62% and 3.30%, respectively) [7]. These results allowed us to evaluate the overall efficacy of the autolysis process.

**Table 4.** Total proteins, sugars, and lipid content of SYR. Results are expressed as mean ± standard deviation. Significant differences are represented by different letters in the same rows for $p < 0.05$. Lowercase letters refer to the whole spent yeast samples and uppercase letters refer to supernatant samples.

| Sample | Protein (% $_{DW}$) | Total Sugars (% $_{DW}$) | Total Lipids (% $_{DW}$) |
|---|---|---|---|
| Untreated whole spent yeast | 40.6 ± 0.01 [a] | 14.52 ± 0.36 [a] | 4.60 ± 0.22 [a] |
| Autolyzed whole spent yeast | 37.9 ± 0.04 [b] | 16.27 ± 1.01 [a] | 4.68 ± 0.18 [a] |
| Untreated supernatant | 41.9 ± 0.08 [A] | 5.92 ± 0.68 [A] | ND |
| Autolyzed supernatant | 45.2 ± 0.23 [B] | 6.30 ± 0.95 [A] | ND |

ND—not determined; DW—dry weight.

In the case of whole SYR, no significant differences were expected between the non-autolyzed and autolyzed samples. However, this was not verified for the protein content parameter, where the amount of protein in the untreated whole yeast sample was found to be higher than that in the autolyzed yeast sample. This difference, despite being lower than 2.7% DW, might be related to the heterogeneity of the yeast samples.

Protein is, without any doubt, the main component in yeast extracts, constituting approximately half of their composition (more than 40% on a dry weight basis for most of the cases [20,43]). The highest protein content was found in the autolyzed samples' supernatants (more than 45% DW), when compared with the non-autolyzed ones (around 42% DW), revealing the release of these components from the intracellular region and indicating an effective cell wall breakdown [41,42], thus confirming the success of the smooth autolysis process. The same results were reported in the literature for extracts from the native spent brewer's yeast [20].

No significant differences were observed with respect to total sugars for both the non-autolyzed and autolyzed samples' supernatants. The total lipid content was only determined in the whole yeast samples since lipids are found on spent yeasts' cellular walls and not in supernatants. The total lipid content of the autolyzed and non-autolyzed SYR samples was around 4.60% DW for both. Regarding the spent yeast supernatant samples and, similarly, the whole yeast samples, autolysis had no significant effect on total sugar content.

4.2.3. Amino Acid Profile

The SYR samples, in general, have a high nutritional value mainly due to their high contents in some essential amino acids. Table 5 shows the amino acid profiles obtained for both the whole yeast and supernatant samples, whether untreated and autolyzed.

The high contents of essential amino acids and the abundance of some of them stand out, thus characterizing this SYR as an excellent material to complement animal diets [27,44]. For instance, lysine and threonine are excellent amino acids to complement an animal cereal diet. Animal diets based on cereals are composed of proteins and are typically deficient in these amino acids [27,44].

**Table 5.** Amino acid profiles of SYR samples. Results are expressed as mean $\pm$ standard deviation. Significant differences are represented by different letters in the same rows for $p < 0.05$. Lowercase letters refer to the whole spent yeast samples and uppercase letters refer to the supernatant samples.

| Amino Acid Profile | Samples | | | |
|---|---|---|---|---|
| | Untreated Whole (% DW) | Autolyzed Whole (% DW) | Untreated Supernatant (% FW) | Autolyzed Supernatant (% FW) |
| Aspartic acid | 10.4 $\pm$ 0.3 a | 10.7 $\pm$ 0.4 a | 5.1 $\pm$ 0.3 A | 8.5 $\pm$ 0.8 B |
| Glutamic acid | 17.8 $\pm$ 0.2 a | 18.4 $\pm$ 0.4 a | 30.5 $\pm$ 1.4 A | 6.4 $\pm$ 0.7 B |
| Cysteine | 1.8 $\pm$ 0.1 a | 1.8 $\pm$ 0.2 a | 1.0 $\pm$ 0.1 A | 0.8 $\pm$ 0.1 A |
| Asparagine | ND | ND | 2.8 $\pm$ 0.1 A | 4.2 $\pm$ 0.1 A |
| Serine | 5.9 $\pm$ 0.0 a | 5.5 $\pm$ 0.1 a | 1.9 $\pm$ 0.1 A | 4.0 $\pm$ 0.1 B |
| Histidine | ND | ND | 2.8 $\pm$ 0.0 A | 2.5 $\pm$ 1.0 A |
| Glutamine | ND | ND | 12.3 $\pm$ 0.7 A | 8.6 $\pm$ 0.8 B |
| Glycine | 4.8 $\pm$ 0.1 a | 5.0 $\pm$ 0.2 a | 1.8 $\pm$ 0.1 A | 2.8 $\pm$ 0.7 A |
| Threonine | 6.4 $\pm$ 0.0 a | 6.9 $\pm$ 0.1 a | 5.7 $\pm$ 0.3 A | 3.6 $\pm$ 0.6 B |
| Arginine | 6.8 $\pm$ 0.2 a | 3.7 $\pm$ 1.0 b | 5.1 $\pm$ 0.3 A | 2.8 $\pm$ 0.2 B |
| Alanine | 10.7 $\pm$ 0.1 a | 9.3 $\pm$ 1.1 b | 14.9 $\pm$ 0.7 A | 12.9 $\pm$ 0.6 B |
| Tyrosine | 3.9 $\pm$ 0.0 a | 3.6 $\pm$ 0.2 a | 1.8 $\pm$ 0.0 A | 3.4 $\pm$ 0.2 B |
| Valine | 5.5 $\pm$ 0.1 a | 6.8 $\pm$ 0.1 b | 5.4 $\pm$ 0.2 A | 7.7 $\pm$ 1.0 B |
| Methionine | 2.0 $\pm$ 0.0 a | 2.1 $\pm$ 0.0 a | 0.8 $\pm$ 0.1 A | 1.4 $\pm$ 0.1 A |
| Tryptophan | ND | ND | 0.3 $\pm$ 0.1 A | 1.3 $\pm$ 0.0 A |
| Phenylalanine | 4.0 $\pm$ 0.1 a | 4.3 $\pm$ 0.1 a | 1.9 $\pm$ 0.0 A | 4.3 $\pm$ 0.1 B |
| Isoleucine | 3.9 $\pm$ 0.2 a | 4.9 $\pm$ 0.0 a | 2.6 $\pm$ 0.0 A | 5.3 $\pm$ 0.2 B |
| Leucine | 6.8 $\pm$ 0.1 a | 7.2 $\pm$ 0.0 a | 2.6 $\pm$ 0.2 A | 9.4 $\pm$ 0.6 B |
| Lysine | 9.3 $\pm$ 0.4 a | 9.7 $\pm$ 0.2 a | 9.5 $\pm$ 0.1 A | 10.2 $\pm$ 1.0 A |

DW—dry weight; FW—fresh weight; ND—not detectable.

As expected, the differences in the amino acid profiles of the whole yeast samples for both untreated and autolyzed ones were not as evident as the ones obtained for the supernatants (free amino acids). Regarding free amino acids, further significant differences could be observed. The glutamic acid and glutamine contents significantly decreased after the induced autolysis process. These are not essential amino acids, and they are closely related in a chemical sense [45].

In contrast, regarding the spent yeast supernatants, a general and significant increase in amino acid concentrations was observed after induced autolysis. These results are in accordance with the ones reported by Podpora et al. [46] in relation to spent brewer's yeast, which stated that along with autolysis time, an increase in free amino acid content occurred. Also, the obtained result is in agreement with the ones reported by [20,38]. Tanguler et al. (2008) [20] and Suphantharika et al. (1997) [38] studied native spent brewer's yeast. These authors asserted that there was a higher breakdown of proteins and peptides at 50 °C when compared to other temperatures. Yeast proteases were inactive at 55 and 60 °C but active at 45 and 50 °C.

Leucine, for example, increased around 7%, while aspartic acid, valine, phenylalanine, isoleucine and serine increased more than 2% after the induced autolysis process. Some animal studies showed that a supplementation with leucine and/or phenylalanine could effectively improve the intestinal starch digestion of ruminants [47]. Another study stated that serine supplementation provided to laying hens fed on low-crude protein diets enhanced humoral and ileal mucosal immunity and attenuated the ileal inflammation of layers [48].

### 4.2.4. Neutral Sugar and Lipid Profiles

The neutral sugar profile (Table 6) was also determined and analyzed for both whole yeast and supernatant samples. The induced autolysis process carried out using selected combinations of temperature, pH and time conditions promoted a slight but non-significant increase in glucose levels in the autolyzed samples of spent yeast residue.

**Table 6.** Neutral sugar content in the whole spent yeast samples and corresponding supernatants. Results are expressed as mean ± standard deviation. Significant differences are represented by different letters in the same rows for $p < 0.05$. Lowercase letters refer to the whole spent yeast samples.

| Sample | Mannose (% DW) | Glucose (% DW) |
|---|---|---|
| Untreated whole spent yeast | 6.10 ± 0.24 [a] | 8.42 ± 0.12 [a] |
| Autolyzed whole spent yeast | 6.67 ± 0.50 [a] | 9.59 ± 0.51 [a] |
| Untreated supernatant | 2.36 ± 0.31 [a] | 3.56 ± 0.37 [a] |
| Autolyzed supernatant | 2.75 ± 0.47 [a] | 3.55 ± 0.49 [a] |

The lipid profile of the non-autolyzed and autolyzed whole yeast samples is shown in Figure 1. Phospholipids represented the main moiety. The phospholipid contents were 28.62 ± 1.88 g/100 g and 26.95 ± 0.12 g/100 g for the non-autolyzed and autolyzed whole spent yeast samples, respectively. Other relevant compounds were hydrocarbons (25.15 ± 1.20 g/100 g and 25.05 ± 0.19 g/100 g for the non-autolyzed and autolyzed whole spent yeast samples, respectively). These results agree with the fact that the analyzed fraction is related to lipids recovered from the membrane of *S. cerevisiae*. where they exert structural functions. On the other hand, this yeast is a synthetic biotechnology-derived organism, which was engineered to produce terpenes.

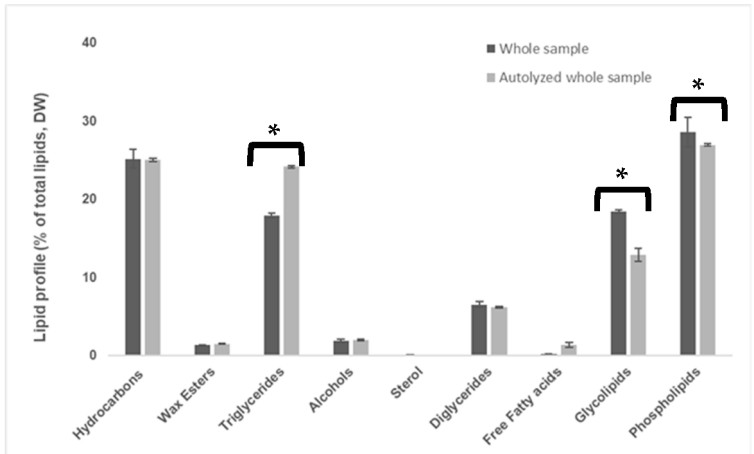

**Figure 1.** Lipid profile of non-autolyzed and autolyzed whole spent yeast residue samples. Results are expressed as mean ± standard deviation. Significant differences are represented by * for $p < 0.05$.

After the induced autolysis process, a decrease in the concentration of glycolipids (18.40 ± 0.21 g/100 g for non-autolyzed vs. 12.88 ± 0.81 g/100 g for autolyzed samples, $p < 0.05$) was observed, followed by increments in the concentrations of triglycerides (17.91 ± 0.31 g/100 g for non-autolyzed vs. 24.15 ± 0.17 g/100 g for autolyzed samples, $p < 0.05$) and free fatty acids (0.17 ± 0.02 g/100 g for non-autolyzed vs. 1.36 ± 0.31 g/100 g for autolyzed whole yeast samples). It is known that alkali conditions can lead to the hydrolysis of glycolipids, releasing sugars and fatty acids. This is supported by the fact that a decrease in glycolipids is accompanied by an increment in free fatty acids as well as in glucose, as shown in Table 6.

4.2.5. Macro- and Micro-Mineral Determination

The macro-mineral and micro-mineral composition of the SYR samples is presented in Figure 2. The mineral profile was determined for the whole yeast samples and their respective supernatants. The amount of minerals found in the whole non-autolyzed and autolyzed samples should be identical. Autolysis should not change the concentration of these compounds when dealing with the entire spent yeast sample. No significant

differences were attained between the autolyzed and non-autolyzed whole spent yeast samples for the majority of minerals analyzed. The only significant difference was found in the aluminum content, which was $1.52 \pm 0.03$ and $0.51 \pm 0.06$ mg L$^{-1}$ for the non-autolyzed and autolyzed supernatants, respectively. This result could be related to the heterogeneity of the samples. As we added NaOH to some samples to raise their pH, sodium was excluded from this analysis.

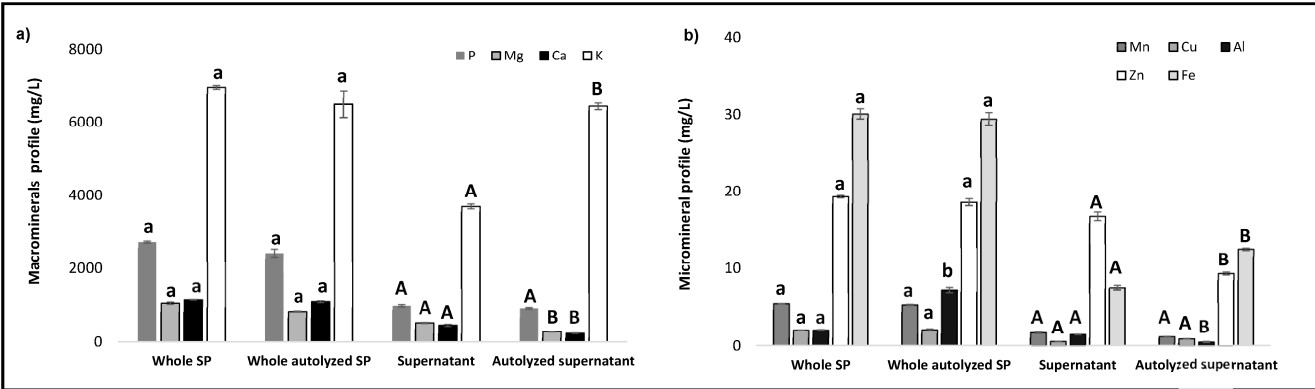

**Figure 2.** Macro- (**a**) and micro-minerals (**b**) contents of non-autolyzed and autolyzed whole spent yeast residue and respective supernatants. Results are expressed as mean $\pm$ standard deviation. Significant differences are represented by different letters above the bars for $p < 0.05$. Lowercase letters refer to the whole spent yeast samples and uppercase letters to the supernatant samples.

Similar to native brewer's spent yeast, potassium is one of the macro-minerals (Figure 2a) which is present in greater amounts [49]. This mineral, together with sodium, plays an important role in the regulation of the cell acid–base balance and water retention, and is essential for ribosomal protein synthesis [49]. Our results showed that the autolysis process induced a significant increase in this mineral in the samples' supernatants.

Regarding micro-minerals (Figure 2b), the results were also similar to what happens for native brewer's spent yeast, with iron and zinc being the prevalent elements. When comparing the untreated and autolyzed samples' supernatants, it is possible to observe that the induced autolysis process decreased the zinc concentration while it slightly increased the iron concentration (Figure 2b) [50]. In other studies, the mineral zinc was found at 30% in yeast cell wall's mannoprotein fractions, at 56% in the vacuole, at 5% in the cytosol, and the rest in other organelles [43]. This fact suggests that the conditions applied during autolysis were not sufficient to allow the release of this mineral to the yeast extract. If the objective was to produce a zinc-enriched yeast extract, the method for the breakdown of cell wall should be directed toward this goal and carried out with the use of cell wall-cleaving β-1,3-glucanase [50].

## 5. Overall Effect of Induced Autolysis Process

As shown in Figure 3, it is possible to observe the overall impact of the induced autolysis process on both the whole yeast and supernatant samples (Figure 3a,b, respectively). In accordance with what has been discussed in the above sections, the overall difference observed after the induced autolysis process is that the whole SYR samples are not as clustered as the supernatants. This means that, although differences between the autolyzed and non-autolyzed samples exist, they are more evident in the case of the supernatants, and more tenuous and less standardized in the case of the whole yeast samples (Figure 3).

For these samples, the important features identified by the analysis are related to the lipid profile, particularly with the free fatty acid content. These parameters appear with more intensity in the whole SYR samples, when compared with the autolyzed ones.

With respect to the sample supernatant analysis (Figure 3b), it is possible to identify a color pattern that unambiguously separates the untreated and autolyzed samples. In these

samples, the highest incidence of the majority of the parameters analyzed is found in the autolyzed samples. This means that these parameters are present in higher levels in the autolyzed supernatants, when compared with the untreated ones, with the exception of minerals and glutamic acid contents.

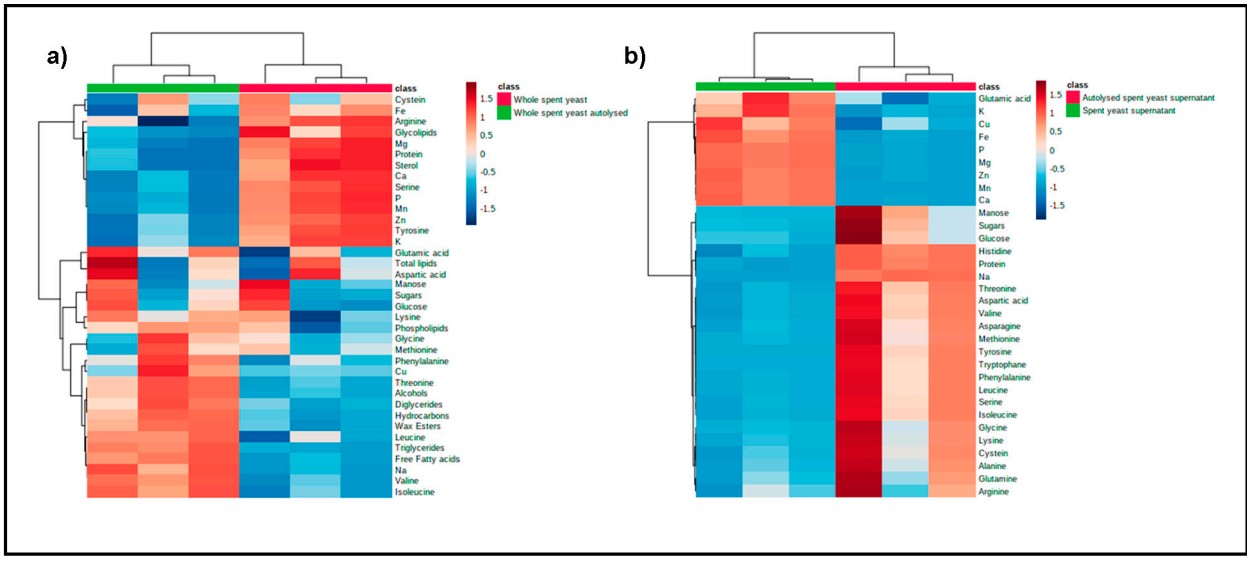

**Figure 3.** Heatmap of the hierarchical clustering analysis generated from SYR characterization data, reflecting the effect of the induced autolysis process on the whole yeast residue samples (**a**) and respective supernatants (**b**).

## 6. General Conclusions

This case study shows the possibility of selecting an autolysis process as a fast and low-cost technology to attain products from a synthetic biotechnology-derived SYR, which are rich in high-value nutritional compounds; this process can be targeted to commercial applications.

The production of complex biomolecules through precision fermentation constitutes a sustainable and innovative practice. This technology allows the production of a myriad of biomolecules using renewable resources such as sugarcane, while preventing the depletion of natural resources. Most of the molecules produced through this technology were previously and are still extracted or captured from nature due to their unique properties. However, to close the sustainability loop and effectively contribute to the establishment of a circular economy, efforts should be made to valorize the increasing amounts of residues produced during these fermentation processes.

The work presented herein clearly shows that the studied SYR derived from synthetic biotechnology is a rich source of amino acids, minerals and other components, with potential for valorization. The determination of the supernatants' dry weight revealed that the induced autolysis time, when at 50 °C, can be reduced from four to two hours if pH is raised from 5 to 8 (18.20% dry weight for 4 h and 18.70% dry weight for 2 h at a higher pH).

The impact of the selected induced autolysis process on the composition of this SYR, in both its whole yeast and respective supernatants, was characterized. From the overall analysis of the autolyzed supernatants, it is possible to attain an improved residue, with a generally higher availability of the compounds of interest. The highest positive impact of the selected autolysis process was observed for the sample's supernatants in terms of proteins, free amino acids and mineral contents.

An increase in leucine (around 7%), aspartic acid, valine, phenylalanine, isoleucine, and serine (approximately 2%) was observed in the autolyzed samples, when compared with the untreated ones. Also, regarding minerals, the autolysis process allowed us to obtain significantly higher amounts of potassium in the treated supernatants.

These parameters are of utmost importance in the animal feed supplementation field. SYRs derived from precision fermentation platforms are rich sources of valuable components, and induced autolysis contributes to their bioavailability, increasing their potential for the development of different applications. Also, the autolyzed whole biomass may be bulked for further processing in biorefineries, aiming at producing new bioproducts that are capable of replacing some of the currently used oil-based products. For this purpose, knowledge must be established for such innovative waste streams since it will allow the development of the most suitable valorization strategies for these new residues.

**Author Contributions:** Conceptualization, J.F.F. and A.L.A.; methodology, J.F.F. and D.A.R.; software, T.D.; validation, M.E.P., L.M.R.-A. and A.L.A.; investigation, J.F.F., D.A.R., T.D., L.L.P. and S.S.M.P.V.; writing—original draft preparation, J.F.F.; writing—review and editing, J.F.F., T.D., M.E.P. and A.L.A.; supervision, M.E.P. and A.L.A. All authors have read and agreed to the published version of the manuscript.

**Funding:** This work was supported by Amyris Bio Products Portugal Unipessoal Lda and Escola Superior de Biotecnologia—Universidade Católica Portuguesa through the Alchemy project—Capturing high value from industrial fermentation bioproducts (POCI-01-0247-FEDER-027578).

**Institutional Review Board Statement:** Not applicable.

**Informed Consent Statement:** Not applicable.

**Data Availability Statement:** The data presented in this study are available on request from the corresponding author. The data are not publicly available due to confidentiality agreements.

**Conflicts of Interest:** The authors declare no conflict of interest.

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
