# Peer review of "Induced Autolysis of Engineered Yeast Residue as a Means to Simplify Downstream Processing for Valorization—A Case Study"

_fermentation, doi:10.3390/fermentation9070673_

Round 1

Reviewer 1 Report (Previous Reviewer 2)

Dear Editor-in-Chief

The authors have improved the paper for potential publication

Minor editing of English language required

Author Response

 We would like to thank you for your opinions and suggestions. The manuscript was improved significantly. All the insertions are in blue font, along the manuscript.

Reviewer 2 Report (New Reviewer)

The manuscript has scientific merit. However, the novelty must be clarified.

What do the authors think to be the main contribution of their work to the knowledge 

The authors must bring a comparison section between their main results and those present in the literature.

Author Response

Following your kind message, please find uploaded the document with the answers to your comments. All insertions are in blue font, along the manuscript. We would like to thank you for your opinions and suggestions. They have surely contributed to improve the manuscript significantly.

Reviewer 3 Report (New Reviewer)

The authors of this manuscript intend to publish as a case study a research study whose results are neither interesting nor significant due to the small effect observed when applying the established treatments. Although in some cases the differences found are statistically significant, such differences are of no practical significance.

The hypothesis is that the chemical composition of the yeast used in this study is different from Native S. cerevisiae but a comparison between both yeast is missing in the paper. On the other hand data on the content of the different compounds in the yeast cell before and after autolysis are also missing. This data will be more informative to understand the autolysis process that the content of the whole spent yeast and whole autolyzed spent yeast.

Other problems detected in the paper are:

The abstract should be content a summary of the results rather than a long justification of the papers

The section autolyses process overview should be removed. This information is too detailed to be included in a research paper. Furthermore, the title of the section may result confuse according to the content. Autolysis is the shelf-degradation of the yeast by its own enzymes but the section mainly refers to different techniques that cause the mechanical destruction of the cells such as US or HPP.

Results are not discussed and scarcely compared with results obtained by other authors with native S. cerevisiae or other yeasts. A more deeply discussion of the results should be required

Conclusions presents a summary of the introduction a justification of the study rather that conclusion derived of the results obtained in the experimental section.

no comments

Author Response

Following your kind message, please find uploaded the document with the answers to your comments. All insertions are in blue font, along the manuscript. We would like to thank you for your opinions and suggestions. They have surely contributed to improve the manuscript significantly.

This manuscript is a resubmission of an earlier submission. The following is a list of the peer review reports and author responses from that submission.

Round 1

Reviewer 1 Report

Dear Editor,

In general, some interesting results were provided here but the quality, from my point of view, may not be contributed to the Journal of fermentation. The topic is hot and significant; however, the present work only analyze the influences of processing conditions, which is already well-known from the previous literatures. The author did not introduce the facing problems of this field in the Introduction, and the novelty of this work is insufficient. In addition, there is also some problems in the academic expression, such as the Sec. 2 should be merged into Sec.1. It looks more like a review rather than an article if just based on the Sec. 1 and 2. The technical analysis on the target products is also simple. As a consequence, I am sorry but I have to offer a “Reject” for this work, and I hope the above comments can help for the further submission.

Not.

Reviewer 2 Report

After carefully evaluating this manuscript, the authors strongly encouraged to clarify the novelty and importance of this work, I have doubts about whether the implementation of the work has led to meaningful results.

Most parts seem a review article mixed with some experiments

All sections especially "Abstract" needs to improve

Lack of statistical explanation in both the text and legends (error bars)

Space between values and number

The quality of figures needs to improve 

As a non-native speaker, I found the manuscript easy to read and understand. However, there are some grammatical errors and in some instances, the phrasing need to improve